# A Validation System for Selection of Bacteriophages against Shiga Toxin-Producing *Escherichia coli* Contamination

**DOI:** 10.3390/toxins13090644

**Published:** 2021-09-11

**Authors:** Agnieszka Necel, Sylwia Bloch, Bożena Nejman-Faleńczyk, Aleksandra Dydecka, Gracja Topka-Bielecka, Alicja Węgrzyn, Grzegorz Węgrzyn

**Affiliations:** 1Department of Molecular Biology, Faculty of Biology, University of Gdansk, Wita Stwosza 59, 80-308 Gdansk, Poland; agnieszka.necel@phdstud.ug.edu.pl (A.N.); sylwia.bloch@ug.edu.pl (S.B.); bozena.nejman-falenczyk@ug.edu.pl (B.N.-F.); aleksandra.dydecka@ug.edu.pl (A.D.); gracja.topka@phdstud.ug.edu.pl (G.T.-B.); 2Laboratory of Phage Therapy, Institute of Biochemistry and Biophysics, Polish Academy of Sciences, Kładki 24, 80-822 Gdansk, Poland; alicja.wegrzyn@ug.edu.pl

**Keywords:** Shiga toxin-producing *Escherichia coli*, bacteriophage, food protection

## Abstract

Shiga toxin-producing *Escherichia coli* (STEC) can cause severe infections in humans, leading to serious diseases and dangerous complications, such as hemolytic-uremic syndrome. Although cattle are a major reservoir of STEC, the most commonly occurring source of human infections are food products (e.g., vegetables) contaminated with cow feces (often due to the use of natural fertilizers in agriculture). Since the use of antibiotics against STEC is controversial, other methods for protection of food against contaminations by these bacteria are required. Here, we propose a validation system for selection of bacteriophages against STEC contamination. As a model system, we have employed a STEC-specific bacteriophage vB_Eco4M-7 and the *E. coli* O157:H7 strain no. 86-24, bearing Shiga toxin-converting prophage ST2-8624 (Δ*stx2*::*cat gfp*). When these bacteria were administered on the surface of sliced cucumber (as a model vegetable), significant decrease in number viable *E. coli* cells was observed after 6 h of incubation. No toxicity of vB_Eco4M-7 against mammalian cells (using the Balb/3T3 cell line as a model) was detected. A rapid decrease of optical density of STEC culture was demonstrated following addition of a vB_Eco4M-7 lysate. However, longer incubation of susceptible bacteria with this bacteriophage resulted in the appearance of phage-resistant cells which predominated in the culture after 24 h incubation. Interestingly, efficiency of selection of bacteria resistant to vB_Eco4M-7 was higher at higher multiplicity of infection (MOI); the highest efficiency was evident at MOI 10, while the lowest occurred at MOI 0.001. A similar phenomenon of selection of the phage-resistant bacteria was also observed in the experiment with the STEC-contaminated cucumber after 24 h incubation with phage lysate. On the other hand, bacteriophage vB_Eco4M-7 could efficiently develop in host bacterial cells, giving plaques at similar efficiency of plating at 37, 25 and 12 °C, indicating that it can destroy STEC cells at the range of temperatures commonly used for vegetable short-term storage. These results indicate that bacteriophage vB_Eco4M-7 may be considered for its use in food protection against STEC contamination; however, caution should be taken due to the phenomenon of the appearance of phage-resistant bacteria.

## 1. Introduction

The vast majority of *Escherichia coli* strains are commensals living in the gut of animals, including humans. They belong to natural intestinal microbiota and are harmless to their hosts under healthy conditions. However, specific strains called Shiga toxin-producing *E. coli* (STEC) are human pathogens [1]. Infection with STEC often causes bloody diarrhea and various complications, including hemolytic uremic syndrome which reveals relatively high morbidity and mortality (10%–20% cases) [2,3]. Shiga toxins are major virulence factors of STEC. They are AB_5_-type protein toxins, consisting of two types of subunits, A and B. The B subunit is responsible for recognizing Gb3 and/or Gb4 receptor in the cell membrane which is the first step of the entrance of the toxin into an eukaryotic cell. Following endocytosis and retro-translocation from Golgi apparatus to endoplasmic reticulum, the A subunit is liberated into the cytoplasm. Then, it acts as a specific enzyme causing removal of one of adenine residues in 28S rRNA which inactivates ribosome function [4].

The *stx* genes, coding for Shiga toxin subunits, are located in the genomes of bacteriophages, called Shiga toxin-converting phages or Stx phages, which occur in STEC cells in the form of prophages [5]. In *E. coli* cells lysogenic with Stx phages, most phage genes are not expressed; this also concerns *stx* genes. However, under stress conditions, Stx prophage is induced which is followed by excision of phage DNA, its replication, and expression of phage genes, including *stx* genes. Progeny virions are formed and released due to host cell lysis. Since Shiga toxins are produced simultaneously, they are liberated together with newly assembled phage particles [5].

Various factors may cause induction of Stx prophages, some mainly in laboratories, such as UV irradiation or treatment with DNA-binding antibiotics, but others, such a oxidative stress, occurring naturally [6]. High variability of Stx phages has been demonstrated, and these viruses reveal differential virion morphology, regulation of specific genes expression, and even DNA replication modules [7,8]. Nevertheless, the common features of Stx phages are the presence of *stx* genes in their genomes, lysogenization of host cells, and effective expression of these genes upon prophage induction [9].

STEC strains are considered dangerous pathogens not only because of severe symptoms caused by infection with these bacteria, but also due to difficulties in treatment. Various antibiotics are also potent inducers of prophages. Hence, administration of antibiotics to a STEC-infected patient may result in massive production of Shiga toxin and severe effects of the toxin action. This is why the use of antibiotics in treatment of STEC infections is controversial, and such a treatment is forbidden in some countries [10,11]. Therefore, it is crucial to develop novel methods for both effective treatment of STEC-infected patients and prevention of STEC infections [12].

There are different ways of infecting humans by STEC, but the vast majority of reported cases of food poisoning indicate vegetables and beef as the sources of pathogenic bacteria [13]. The reason for this is that cattle are the main reservoir of STEC [14]. Gb3 receptor is absent in bovine vascular cells, thus Shiga toxins cannot be effectively transmitted inside such cells [15,16], making these animals resistant to the toxins. Many local epidemic events caused by STEC were reported to be caused by consumption of vegetables contaminated with these bacteria, most probably due to the use of natural fertilizers in agriculture procedures [17]. Therefore, one possible way to prevent STEC infection might be protection of food, specifically vegetables, against STEC contamination.

One possible method of elimination of pathogenic bacteria is the use of bacteriophages specific to selected strains. Recently, we have isolated and characterized bacteriophage vB_Eco4M-7 which infects many *E. coli* strains belonging to the O157 serotype [18]. Since most known STEC strains belong to this serotype [13,19,20], such a phage might be considered as a potential agent allowing protection of food against contamination with such bacteria. This proposal may be corroborated by results of studies that indicated inability of vB_Eco4M-7 to lysogenize host cells, and absence of any genes encoding known toxins and virulence factors in its genome [18]. Therefore, in this work, we aimed to test potential usefulness of vB_Eco4M-7 in food protection against STEC contamination. We have developed a validation system for selection of bacteriophages against STEC contamination which was used for testing vB_Eco4M-7 and can be employed for other phages.

## 2. Results

To test the ability of phage vB_Eco4M-7 to eliminate STEC from food, we have employed an experimental system consisting of a model vegetable-cucumber, and model STEC bacteria-the *E. coli* O157:H7 strain no. 86–24, bearing Shiga toxin-converting prophage ST2-8624 (Δ*stx2::cat gfp*) [20,21]. As shown in Figure 1, cucumber was cut into slices, and following UV sterilization bacteria were spotted at the amount of 10^8^ colony forming units (CFU) per ml. Then, phage lysate was applied to various multiplicity of infection (MOI), and after 6 and 24 h incubation the number of survived bacterial cells was determined.

Application of the lysate of phage vB_Eco4M-7 resulted in a significant decrease in the number of living bacteria after 6 h incubation (Figure 2). Interestingly, efficiency of the anti-bacterial activity of the phage lysate was observed for each tested MOI value, ranging from 0.0001 to 10 (Figure 2). Longer incubation (24 h) resulted in bacterial re-growth and less pronounced antibacterial activity of the phage lysate, though there were still significant differences in the titers of the STEC strain treated and non-treated with the bacteriophage (Figure 2). Interestingly, vB_Eco4M-7 was more efficient in killing STEC when used at lower MOI values (0.0001–0.01).

Knowing that application of bacteriophage vB_Eco4M-7 on food may efficiently decrease the number of STEC cells, one might assume that this virus can be potentially used in food protection procedures. However, such an application requires testing the safety of phage lysate. To assess putative effects of vB_Eco4M-7, we have investigated the response of mammalian cells (the Balb/3T3 line) to the presence of this bacteriophage. No decrease in viability of these cells was found after 24 h incubation with phage vB_Eco4M-7 lysate (10^9^ PFU/mL) (Figure 3a). Moreover, Balb/3T3 cells had normal morphology as assessed microscopically (Figure 3b). Contrary to phage vB_Eco4M-7 incubation of Balb/3T3 cells with the *E. coli* O157:H7 (ST2-8624) strain for 24 h resulted in a drastic decrease of viability of mammalian cells (Figure 3a). Such a deleterious effect of the STEC strain on Balb/3T3 cells was evident also in the presence of phage vB_Eco4M-7 (Figure 3a), despite decreased number of bacterial cells (Figure 3c) resulting from the action of phage particles remaining at a similar level (Figure 3d) after 24 h of incubation. When bacterial survivors were analyzed, it was found that the majority of cells were mutants resistant to bacteriophage vB_Eco4M-7 (Figure 3e).

Since growth of the STEC strain was affected differentially by bacteriophage vB_Eco4M-7 in the food protection experiment, depending on the number of phage particles applied to cucumber slices, we investigated effects of this phage on *E. coli* O157:H7 (ST2-8624) bacteria cultures in a liquid medium following infection at various MOI. When bacterial culture was infected with vB_Eco4M-7, we observed a rapid decrease of culture density during first 4 h at MOI 0.1, 1 and 10, indicating effective killing of sensitive bacteria (Figure 4). As expected, under low MOI value (0.01, 0.001, and 0.0001) conditions, culture density was slower, and to a less extent. However, after 4 h of incubation, densities of cultures infected at MOI 0.1, 1 and 10 started to increase, perhaps representing selection of phage-resistant mutants and their outgrowth (Figure 4). At times 12 and 24 h, densities of infected cultures were indistinguishable from that containing uninfected bacteria. This phenomenon was not observed in cultures infected at MOI 0.01, 0.001 and 0.0001, where growth of bacterial culture was significantly slower at later times of incubation (Figure 4). These results suggest significantly more efficient selection of phage-resistant bacteria when infection occurs at high MOI than under conditions of low MOI.

Under the same experimental conditions (infections at different MOI of the liquid culture of *E. coli* O157:H7 (ST2-8624) with bacteriophage vB_Eco4M-7), efficient bacteriophage lytic development was observed at each tested MOI (from 0.0001 to 10), as revealed by a rapid increase of the phage titer (Figure 5). This indicates that vB_Eco4M-7 can multiply effectively at any phage-bacteria ratio.

To assess efficiency of appearance of vB_Eco4M-7-resistant mutants of *E. coli* O157:H7 (ST2-8624) after 24 h of infection with this phage in a liquid culture, we have estimated number of survivors, and those among them resistant to the phage fraction of cells. The experiments were performed at different MOI values. Perhaps surprisingly, the higher number of bacterial survivors was determined in experiments with the highest tested MOI. On the contrary, the most efficient killing of bacteria occurred at MOI of 0.001 (Figure 6). In all cases, the majority of survivors were bacteria resistant to phage vB_Eco4M-7 (Figure 6), confirming that the growth of *E. coli* O157:H7 (ST2-8624) cultures infected by this phage, observed after initial decrease in the bacterial culture density (as demonstrated in Figure 4), was due to selection of phage-resistant mutants. Interestingly, the significant difference in the number of survived and resistant bacteria was noticed at MOI 0.1 (Figure 6). These results corroborate the hypothesis that incubation of bacteria susceptible to phage infection with such a phage at high MOI provide conditions with a strong selection pressure, leading to isolation of the resistant mutants, which can subsequently multiply in the culture containing a relatively high density of phage virions.

Most of our experiments were conducted under laboratory conditions, with incubations at 37 °C. However, short-term storage of vegetables is usually conducted at lower temperatures, like those occurring in stalls. Therefore, we have tested if vB_Eco4M-7 can develop in *E. coli* O157:H7 (ST2-8624) not only at 37 °C, but also at 25 °C and 12 °C, which better mimic the above described conditions. We observed that bacteriophage vB_Eco4M-7 was able to form plaques at all tested temperatures (37, 25, and 12 °C), indicating that it can develop and destroy STEC cells under temperature conditions likely occurring during vegetable storage (Figure 7).

## 3. Discussion

Due to the severity of infections caused by STEC and a lack of efficient drugs against these bacteria arising from the fact that the use of antibiotics often provokes enhanced production of Shiga toxins, development of novel therapeutic methods is an urgent need [9]. Phage therapy is one possible alternative for treatment of bacterial diseases, as bacteriophages are able to develop lytically in sensitive hosts, thus they can efficiently eliminate these microorganisms [22]. Apart from potential application of bacteriophages as therapeutic agents, another way of using these viruses to protect food against pathogenic bacteria is possible. This applies especially to foodborne pathogens whose transmission to humans occurs mostly through consumption of contaminated meals or drinks [17]. Since the majority of STEC-caused infections arise as a result of food contamination, a potential use of phages to prevent such events appears reasonable. However, to properly assess usefulness of specific bacteriophages in such an application, it is necessary to develop a validation system for selection of bacteriophages against STEC contamination. In this report, we present such a new system which can be useful in choosing phages effective in preventing foodborne STEC infections.

When developing the validation system, we assessed conditions which are tighter and more endangered by bacterial infection than under field conditions of transportation and storage of vegetables. Therefore, if a tested bacteriophage can eliminate pathogenic bacteria under such experimental conditions, it should also be effective in practice. Such a system allows avoidance of false positive results, where, despite high efficiency of the procedure under laboratory conditions, it might be useless under conditions of real transportation and storage. To achieve this, we have used sliced cucumbers instead of whole vegetables. Colonization of whole cucumbers is considerably less likely than that of sliced, as the peel is a natural barrier, and is significantly more difficult for bacteria to attach to. A sliced vegetable has different moisture and consistency which make it considerably more susceptible for colonization by bacteria than an intact one. On the other hand, we cannot exclude partial damage of the peel during transportation and/or storage, thus, a possibility of bacterial contamination of the internal parts of cucumber is not unrealistic. Moreover, despite procedures for washing of cucumbers before shipping, and chlorination of washing water, contamination with STEC cannot be excluded, especially if such procedures are not strictly followed. In fact, examples of such failures which caused STEC infections of humans have been reported [13]. It is worth being reminding that for effective infection and subsequent severe symptoms in patients, a low dose of STEC (below 100 bacterial cells) is sufficient [1,2,3,4,5,6]. Considering these facts, if cucumber surface is contaminated by feces containing STEC, peeling in a kitchen may cause a transfer of bacterial cells inside the vegetable, making growth conditions similar to those used in our experiments. We have also used a number of bacterial cells significantly higher than under natural conditions, assuming that if a tested bacteriophage is effective in eliminating its host in such an experiment, it should also be effective under field conditions. Finally, temperature of 37 °C is optimal for STEC growth, thus, the use of this temperature provided fully permissive conditions for the investigated bacteria. On the other hand, we have also tested the ability of a bacteriophage to propagate and to lyse host cells at lower temperatures, 12 and 25 °C, resembling conditions occurring during actual transportation, storage and selling. A summary of the validation system is presented in Figure 1.

Previously, we have reported isolation and characterization of a newly discovered bacteriophage, vB_Eco4M-7 [18]. This phage was found to infect many *E. coli* O157 strains, including STEC. Genome of vB_Eco4M-7 has been sequenced and no genes coding for known or putative toxins were identified. It is a virulent phage, unable to lysogenize host cells, and revealing effective propagation in host cells. Importantly, vB_Eco4M-7 infection of the *E. coli* O157:H7 host bearing a modified ST2-8624 prophage, in which the *stx* genes were replaced with the *gfp* gene, did not cause an increase in the amount of GFP, indicating that expression of Shiga toxin-encoding genes should not be activated in the presence of the tested bacteriophage [18]. Therefore, vB_Eco4M-7 might be a good candidate for either phage therapy or food protection against STEC strains. Here, we tested its potential in food protection, as a model bacteriophage for the proposed validation system.

Our experiments indicated that vB_Eco4M-7 could efficiently decrease the number of bacterial cells of the model STEC strain, in both liquid culture and in the model food protection experimental system (sliced cucumber inoculated with bacteria). This might corroborate the proposal of the use of vB_Eco4M-7 in food protection against STEC. Nevertheless, we found that the phage-resistant mutants can be selected after prolonged incubation of bacteria in the presence of this phage. Intriguingly, the selection of such mutants was more efficient at high MOI (like 1 or 10) than at low MOI (like 0.001 or 0.0001). Such a phenomenon has been reported previously for another phage-bacteria system, phage vB_EfaS-271 and *Enterococcus faecalis* [23]. This might arise from a strong selection pressure under high MOI conditions, where virtually every bacterial cell is infected by bacteriophage, thus only phage-resistant mutants may survive, and then propagate in the presence of the phage. Under low MOI conditions, only a small fraction of cells is infected, thus the selection pressure is weaker, resulting in growth competition between cells sensitive and resistant to phages. Since most phage-resistant bacterial mutants are often partially deficient in some cellular functions (as a result of mutation/s), the wild-type cells win the competition and outgrow mutants, as long as phage population becomes abundant enough to infect every cell in the population. This can result in slower selection of phage-resistant mutants.

The question remains whether bacteriophage vB_Eco4M-7 can be efficiently used in food protection against STEC under conditions of real food storage. Most of our experiments were performed in a laboratory, with a high number of bacterial cells, and at 37 °C. This was convenient for assessment of general mechanisms and processes accruing during phage-bacteria interactions. However, occurrence of as high a number of STEC bacteria as 10^8^ cells on vegetable surface is unlikely under field conditions. Results indicating that even in these experiments an application of phage vB_Eco4M-7 lysate could significantly reduce number of living bacteria suggest its potential efficiency in food protection. It is also worth noting that phage-resistant mutants occur at a frequency of about 10^−5^ [24], thus if the actual number of STEC cells on food products is lower than 10^5^, then selection of vB_Eco4M-7-resistant mutants might be inefficient. Although storage of vegetables at 37 °C occurs rarely, while lower temperatures are usually employed, the fact that vB_Eco4M-7 copes with bacteria in optimal growth conditions proves its high anti-bacterial properties. This also makes the proposed protection method potentially more effective under field conditions than under laboratory conditions, since bacterial growth is quicker in the latter case. Apart from that, phage vB_Eco4M-7 effectively infected bacteria at lower temperatures, 12 °C and 25 °C (Figure 7), resembling conditions of vegetable storage and those occurring in food stores or stalls.

Previous studies indicated that bacteriophage vB_Eco4M-7 is relatively resistant to environmental conditions [25]. It is highly stable during storage at such diverse temperatures as −20 °C and 40 °C, and even a short time (5 min) incubation at 95 °C resulted in its survival at a level of about 20% [25]. Moreover, vB_Eco4M-7 virions retained their infectivity after storage at pH 4 and 10, as well as following incubation in 0.09% SDS, 50% DMSO or chloroform [25]. This phage was also demonstrated to be resistant to various disinfectants, including 10% soap, 10% dish soap, Line Antibacterial 70, and Virusolve [18]. These properties are favorable in the light of the potential use of phage vB_Eco4M-7 in the field. However, it remains to be determined how a phage preparation should be applied. It is possible to either pour a phage lysate or spay it over the vegetables. Another option is to dip the vegetables in the solution containing bacteriophage virions. Nevertheless, the most effective way of application of bacteriophages should be identified during specific field studies, to ensure the highest efficiency of food protection against STEC contamination.

## 4. Conclusions

A validation system for selection of bacteriophages against STEC contamination has been developed. Using this system, we obtained results suggesting that application of bacteriophage vB_Eco4M-7 lysate can be considered as a potential method for food protection against STEC. Phage-resistant mutants can be selected, but the selection is more efficient at higher MOI. Nevertheless, selection of such mutants appears unlikely under field conditions, though this suggestion requires further investigations during real food storage.

## 5. Materials and Methods

### 5.1. Bacteria, Media and Growth Conditions

The *E. coli* O157:H7 strain 86-24, bearing Shiga toxin-converting prophage ST2-8624 (Δ*stx2*::*cat gfp*) [20,21] was used in this study as the bacterial host for the lytic phage vB_Eco4M-7 [18,25]. For all experiments, host bacteria were cultured in liquid Luria-Bertani (LB) broth medium (EPRO, Władysławowo, Poland) at 37 °C on a rotary shaker at a rate of 200 rpm. The Petri dishes with LB medium containing 1.5% agar (LA medium; BTL, Łódź, Poland) were incubated at 37 °C for 20 h. The soft top agar containing LB broth (EPRO, Władysławowo, Poland) was prepared with 0.7% agar (LA medium; BTL, Łódź, Poland) for phage plaque confirmation.

### 5.2. Preparation of Phage vB_Eco4M-7 Lysate

The vB_Eco4M-7 lysate was prepared according to the method described in a previous study [18]. Briefly, host bacteria were grown to an OD_600_ of 0.2 at 37 °C. Phage lysate was added at an multiplicity of infection (MOI) of 0.1, and the culture was subsequently incubated at 37 °C for a further 2 h with agitation, until cell lysis occurred. Following bacterial lysis, the vB_Eco4M-7 lysate was treated with 4% chloroform (Chempur, Piekary Śląskie, Poland) for 15 min and cleared of cellular debris by centrifugation (2000× *g*, 10 min, 4 °C). The obtained supernatant was then passed through 0.22-µm-pore-size filters (Corning, New York, NY, USA) and the phage particles were precipitated overnight at 4 °C with 10% polyethylene glycol 8000 (PEG8000; EPRO, Władysławowo, Poland). To recover virions of vB_Eco4M-7, the mixture was centrifuged (8000× *g*, 20 min, 4 °C) and the pellet was suspended in TM buffer (10 mM Tris-HCl, 10 mM MgSO4; pH 7.2). The phage particles were extracted with equal volume of chloroform (Chempur, Piekary Śląskie, Poland) by gentle inversion, and phases were then separated by centrifugation (2000× *g*, 10 min, 4 °C). The upper phase, containing vB_Eco4M-7 particles, was collected and the titer of phage lysate was determined by using the plaque assay described in Section 5.3.

### 5.3. Double Agar Layer Assay

The titer of vB_Eco4M-7 lysate (Plaque Forming Units per 1 mL or PFU/mL) was determined by visualization of plaques on LA plates using the soft agar overlay technique as previously described [18]. Briefly, 1 mL of an overnight *bacterial* culture was mixed with 2 mL of melted top agar and poured onto LA bottom agar. Phage lysate was serially diluted in TM buffer (10 mM Tris-HCl, 10 mM MgSO4; pH 7.2) and then 2.5 µL of each dilution was spotted onto the surface of the top agar. The Petri dishes were incubated at 37 °C overnight and the plaque-forming units were counted.

### 5.4. Bacteriophage Treatment of Experimentally Contaminated Cucumber

Fresh cucumber was purchased from the local supermarket and stored at 4 °C before use. The procedure of application of phage vB_Eco4M-7 onto the surface of cucumber slices, inoculated with host bacteria, is schematically presented in Figure 1. Briefly, cucumber was peeled and cut into 5 mm thick slices with an average diameter of 30 mm. In the next step, the slices were divided into quarters and transferred to sterile Petri dishes. Both sides of the quarters of cucumber slices were UV-treated for 1 h in a bio-safety cabinet to decontaminate indigenous bacteria or phages. Then, 25 µL of prepared *E. coli* O157:H7 (ST2-8624) inoculum (10^8^ CFU/mL) was spotted onto quarters of cucumber slices and dried for 40 min at room temperature under sterile conditions. Next, phage lysate was added onto the surface of each sample to achieve MOI of 10, 1, 0.1, 0.01, 0.001 and 0.0001 (four quarters for one MOI value). Non-phage-infected samples (negative control) were treated with appropriate volume of TM buffer (10 mM Tris-HCl, 10 mM MgSO4; pH 7.2). After 6 or 24 h of incubation at 37 °C, each quarter of cucumber slice was transferred to the sterile tube containing 1 mL of PBS buffer and manually homogenized to release bacteria. The number of *E. coli* survivors per 1 mL of the homogenate was determined as described in Section 5.5.

### 5.5. Bacteria Enumeration

To enumerate the number of bacterial cells per 1 mL of the analyzed suspension (Colony Forming Units per 1 mL or CFU/mL), samples were harvested at the indicated times and serially (10-fold) diluted in 0.85% sodium chloride (Chempur, Piekary Śląskie, Poland). In the next step, 40 μL of each dilution were spread on LB agar plates. After overnight incubation at 37 °C, the colony forming units were counted.

### 5.6. Cytotoxicity Assay

The cytotoxicity of phage vB_Eco4M-7 was determined according to the PN-EN ISO 10993-5:2009, Annex A–the Neutral Red Uptake (NRU) cytotoxicity test by the Animal Research Facility at Medical University in Łódź (Poland), as described previously [23]. The NRU viability assay quantifies the number of viable, uninjured cells after their exposure to tested agent, through reduction of the level of the absorbance. Mouse embryonic fibroblast BALB/c3T3 clone 31 cell line (catalog number: 62485414) was obtained from the American Type Culture Collection (ATCC; Manassas, VA, USA). Briefly, mammalian cells were seeded into 96-well plates at density of 10^4^ cells per well and cultured in Dulbecco’s Modified Eagle’s Medium (Biowest, Riverside, MO, USA) containing 20% newborn calf serum (PAN-Biotech, Aidenbach, Germany), 2 mM glutamine (Biological Industries, Kibbutz Beit-Haemek, Israel), 1% sodium pyruvate (Biological Industries, Kibbutz Beit-Haemek, Israel), 100 units/mL of penicillin, and 100 mg/mL of streptomycin (Biowest, Riverside, MO, USA). After 24 h incubation at 37 °C and 5% of CO_2_, the culture medium was discarded from each well and Balb/c3T3 cells were treated with 200 µL of mixtures composed of culture medium and either vB_Eco4M-7 particles (10^9^ PFU/mL) or *E. coli* O157:H7 (ST2-8624) cells (10^8^ CFU/mL) or both. A positive and negative control consisting of 0.01% Triton X-100 or culture medium, respectively, were also prepared. The 96-well plates containing all tested variants were incubated for 24 h under the conditions described above. On the next day, the mix solution was removed, and mammalian cells were washed with 150 µL of PBS. Then, the NR dye solution (0.005%) was added to each well and plates were incubated in the dark at 37 °C and 5% of CO_2_ for 3 h. After removing the NR dye solution, the cells were washed with 150 µL of PBS and subsequently treated with 100 µL of NR desorb solution (50% ethanol and 1% glacial acid in water) to extract the NR from the cells and form a homogeneous mixture. NR absorption was measured at an optical density of 540 nm (OD_540_) in a microplate reader. The cell viability was presented as a percentage of the control values (cell treated with sterile medium). The cytotoxicity assay was performed in triplicate. In accordance with the PN-EN ISO 10993-5:2009 standard, if the cell viability is below 70%, the tested agent is considered cytotoxic. During this experiment, phage titer (PFU/mL) and the number of bacterial cells per ml (CFU/mL) were also determined according to the procedures described in Section 5.3 and Section 5.5, respectively. Additionally, the appearance of vB_Eco4M-7-resistant bacteria after 24 h was analyzed in accordance with the description presented in Section 5.8.

### 5.7. Microscopic Analyses

To determine the morphological changes and growth inhibition of mouse embryonic fibroblast Balb/c3T3 cells after 24 h of treatment with vB_Eco4M-7 particles and/or *E. coli* O157:H7 (ST2-8624) bacteria, the mammalian cells were photographed under the light microscope with phase contrast (OptaView 7 software, Opta-Tech, Warsaw, Poland).

### 5.8. Development of Host Resistance to Phage Infection

To determine the number of bacterial colonies resistant to phage infection, each bacterial survivor from vB_Eco4M-7-infected cultures (at different MOI) was regrown in LB medium to OD_600_ of 0.1 at 37 °C with shaking. Next, the phage lysate was added to each sample to achieve MOI 0.1. The lack of bacterial cell lysis indicated the emergence of host resistance to vB_Eco4M-7 infection.

### 5.9. Killing Efficacy of vB_Eco4M-7against Host Strain

Lytic ability of phage vB_Eco4M-7 at different MOI was demonstrated using exponential-growth-phase host culture as described earlier [18], with some modifications. Briefly, *E. coli* O157:H7 (ST2-8624) cells were cultured to OD_600_ of 0.1 at 37 °C. The phage stock solution of vB_Eco4M-7 was added to the host at MOI of 10, 1, 0.1, 0.01, 0.001 and 0.0001. Subsequently, phage-infected bacteria were incubated with shaking at 37 °C for 24 h. During this experiment, density of bacterial culture, monitored by OD_600_ measurement, and the number of phage particles per 1 mL of suspension (PFU/mL) were determined at the indicated times. To estimate the PFU/mL, collected samples were diluted in TM buffer (10 mM Tris-HCl, 10 mM MgSO_4_; pH 7.2) and then the mixture was spotted onto a double-layer agar plates. Following overnight incubation at 37 °C, the phage titer was calculated on the basis of visible plaques. Additionally, the number of viable bacterial cells and phage-resistant bacteria after 24 h from the vB_Eco4M-7 infection was also calculated as described in Section 5.5 and Section 5.8, respectively.

### 5.10. Statistical Analysis

Variation among biological replicates was presented as error bars indicating the standard deviation (SD). All data comparisons were made by using Student’s *t*-test. The significance of differences between compared groups are marked by asterisks as follows: *p* < 0.05 (*), *p* < 0.01 (**) or *p* < 0.001 (***).

## Figures and Tables

**Figure 1 toxins-13-00644-f001:**
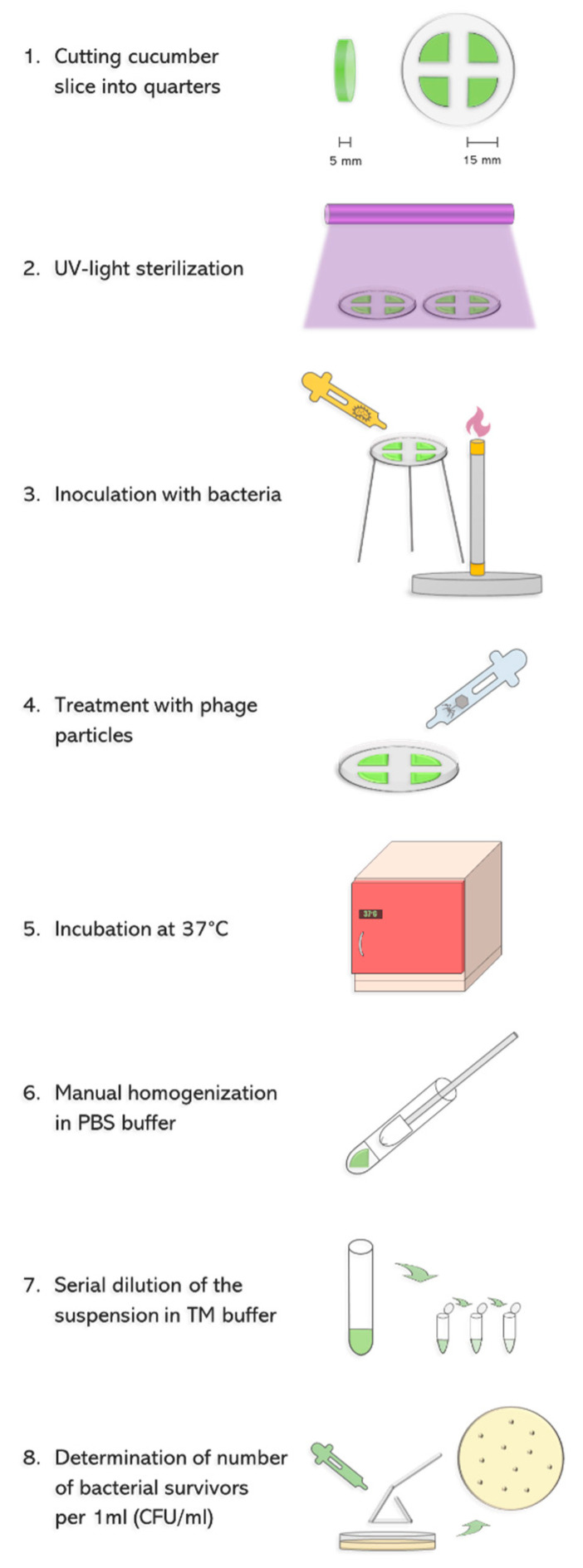
Scheme of experiment devoted to assessment of ability of phage vB_EcoM4-7 to eliminate STEC cells from cucumber slices. See Section 5.4 for details.

**Figure 2 toxins-13-00644-f002:**
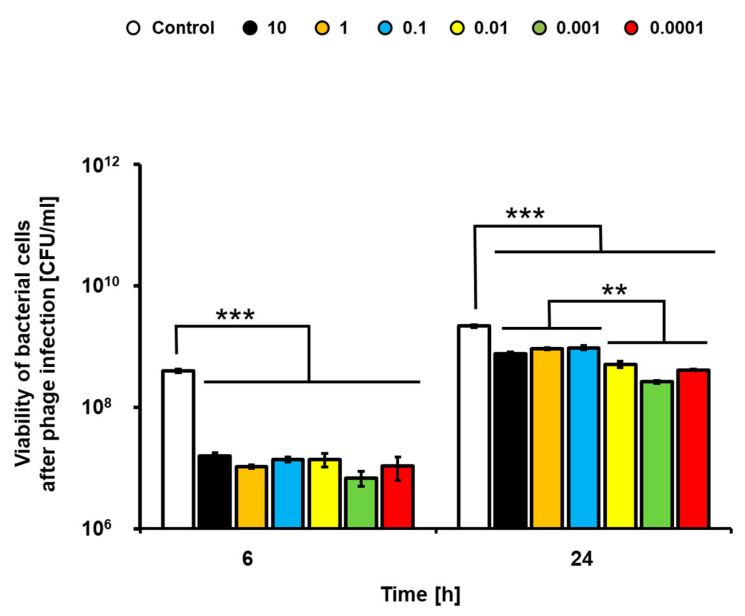
Food application of phage vB_Eco4M-7. Efficiency of phage vB_Eco4M-7 was tested against *E. coli* O157:H7 (ST2-8624) in cucumber slices at different MOI 10 (black columns), 1 (orange columns), 0.1 (blue columns), 0.01 (yellow columns), 0.001 (green columns) and 0.0001 (red columns). Number of bacterial cells per 1 mL (CFU/mL) was determined at the indicated time points. As a negative control (white columns), *E. coli* O157:H7 (ST2-8624) host culture was inoculated with TM buffer instead of the tested virus. Each column represents the mean of three independent experiments, and error bars indicate the standard deviation. Statistical analyses were performed by Student’s *t*-test. Asterisks indicate significant differences between test groups: *p* ≤ 0.01 (**) or *p* ≤ 0.001 (***).

**Figure 3 toxins-13-00644-f003:**
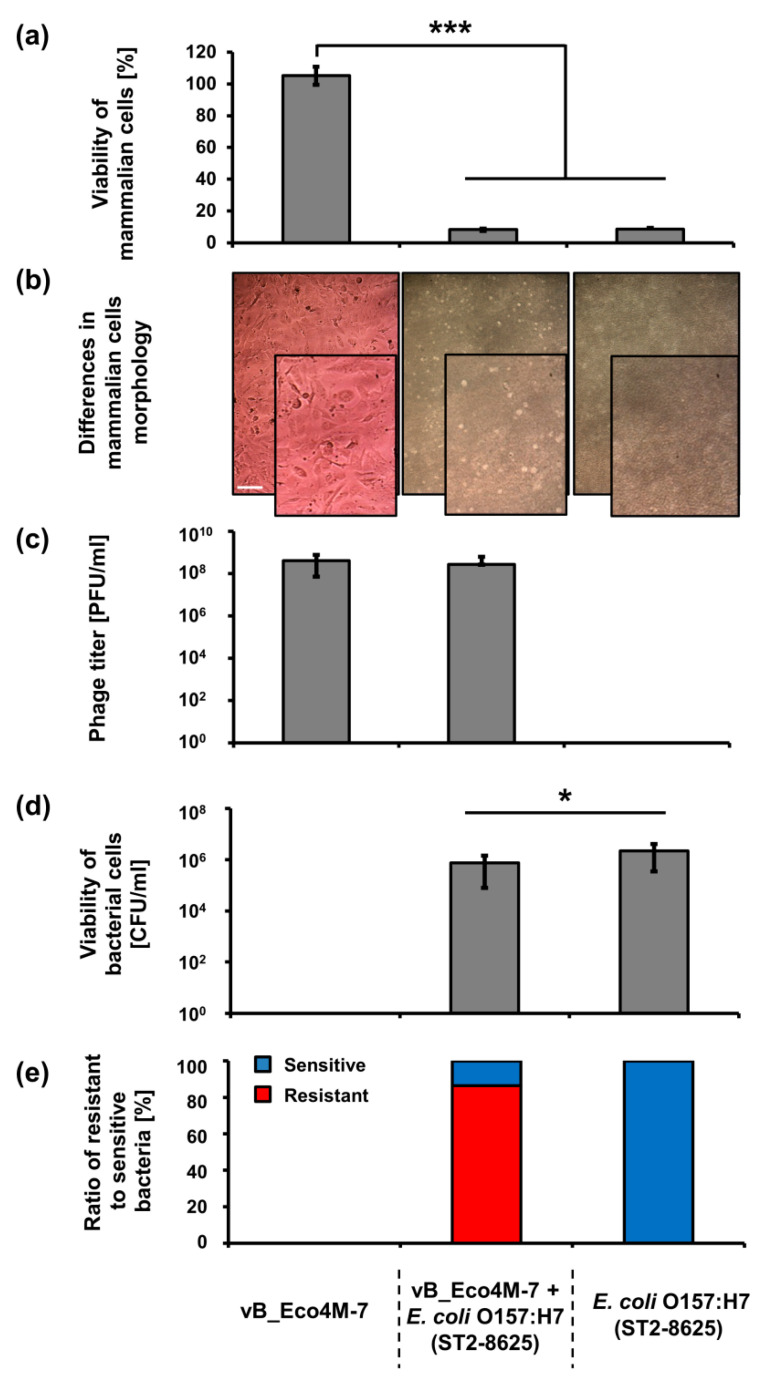
Assessment of cytotoxicity of vB_Eco4M-7 phage particles to mammalian cells and their effects on co-cultured *E. coli* O157:H7 (ST2-8624) bacteria. (**a**) Viability of Balb/3T3 cells exposed to phage vB_Eco4M-7 and/or *E. coli* O157:H7 (ST2-8624) for 24 h. (**b**) Differences in Balb/3T3 cell morphology after treatment with vB_Eco4M-7 phage particles and/or *E. coli* O157:H7 (ST2-8624) bacteria for 24 h. Images were made using light microscopy with phase-contrast. For a better comparison, a selected part of each photo is enlarged (lower-right corner) (**c**) Number of phages per 1 mL (PFU/mL) observed after 24 h treatment with vB_Eco4M-7 suspension added at concentration of 10^9^ PFU/mL or at MOI 10. (**d**) Number of viable *E. coli* O157:H7 (ST2-8624) cells per 1 mL (CFU/mL) quantified after 24 h of phage treatment, and (**e**) percentage ratio of phage-resistant (red) and phage-sensitive (blue) bacteria. Signatures of the X-axis are the same for all panels and are placed at the bottom of this figure for clarity. Due to the nature of the experiments testing only bacterial cells (CFU/mL) or phage particles (PFU/mL), panels **c–****e** lack corresponding columns. Mean values from three independent experiments are shown with error bars indicating SD. Bar represents 100 µm. Statistical analyses were performed using Student’s *t*-test. Significant differences are marked by asterisks (where * indicates *p* < 0.05, and *** indicates *p* < 0.001).

**Figure 4 toxins-13-00644-f004:**
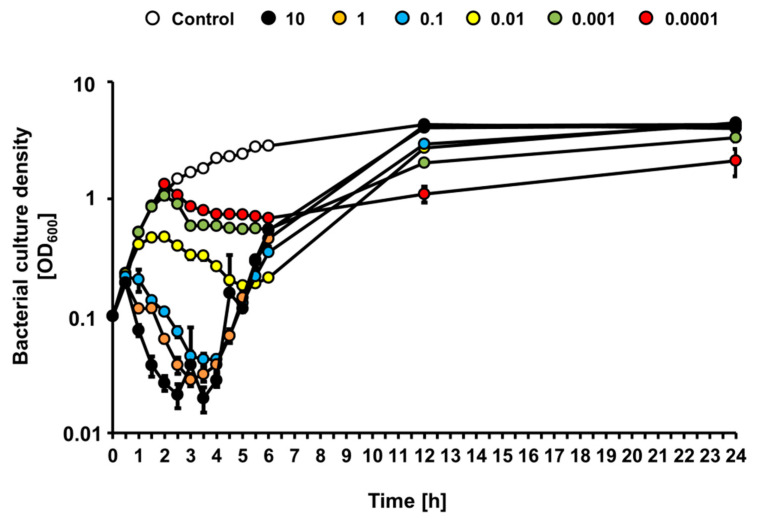
Bacterial culture density (OD_600_) after infection of *E. coli* O157:H7 (ST2-8624) host with phage vB_Eco4M-7 at different MOI 10 (black circles), 1 (orange circles), 0.1 (blue circles), 0.01 (yellow circles), 0.001 (green circles) and 0.0001 (red circles). As a negative control, *E. coli* O157:H7 (ST2–8624) culture was inoculated with LB medium instead of phage vB_Eco4M-7 (white circles). Mean values from three independent experiments are shown with error bars indicating SD. Note that, in some cases, error bars are smaller than size of symbols.

**Figure 5 toxins-13-00644-f005:**
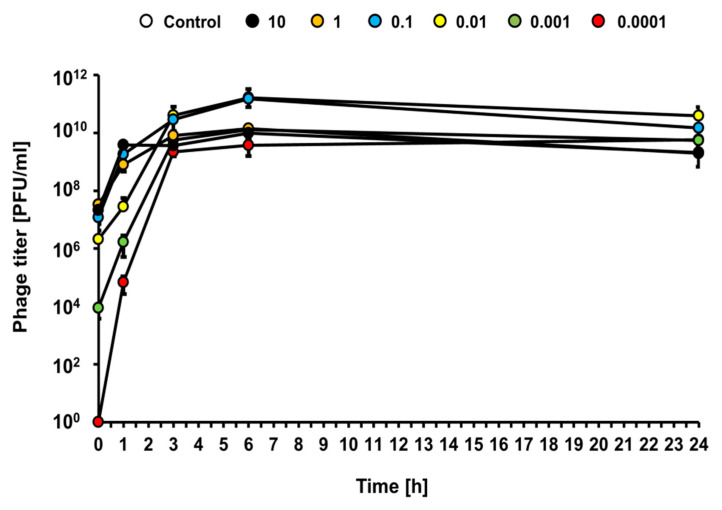
Kinetics of lytic development of phage vB_Eco4M-7 after infection of *E. coli* O157:H7 (ST2-8624) bacteria at different MOI 10 (black circles), 1 (orange circles), 0.1 (blue circles), 0.01 (yellow circles), 0.001 (green circles) and 0.0001 (red circles). The results are presented as the number of plaque forming units per 1 mL (PFU/mL). Mean values from three independent experiments are shown with error bars indicating SD. Note that, in some cases, error bars are smaller than size of symbols.

**Figure 6 toxins-13-00644-f006:**
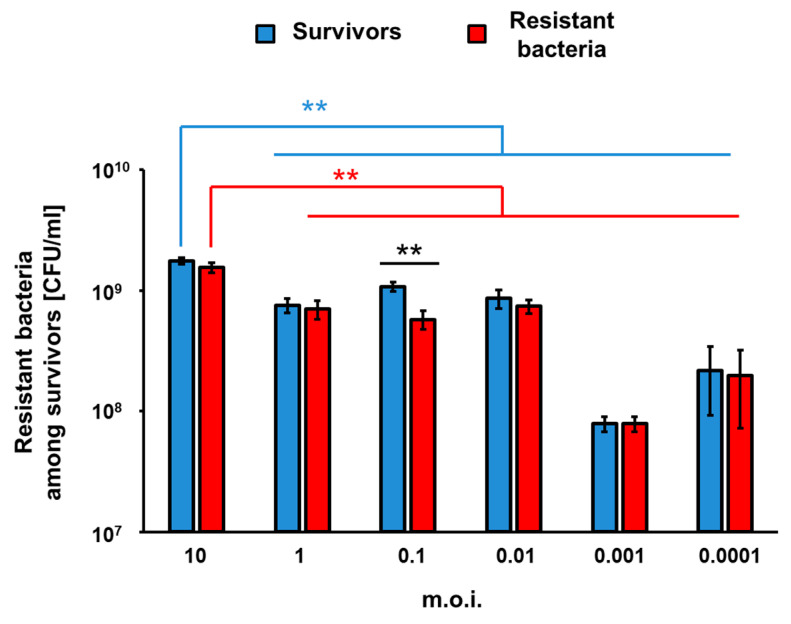
Number of surviving cells (blue columns) and phage-resistant bacterial mutants among survivors (red columns) per 1 ml (CFU/mL) after 24 h from the infection of *E. coli* O157:H7 (ST2-8624) host with phage vB_Eco4M-7 at different MOI (10, 1, 0.1, 0.01, 0.001, 0.0001). Results are presented as mean values ± SD from three biological experiments. Statistical analyses were performed using Student’s *t*-test. Asterisks (**) indicate significant differences (*p* ≤ 0.01) between test groups.

**Figure 7 toxins-13-00644-f007:**
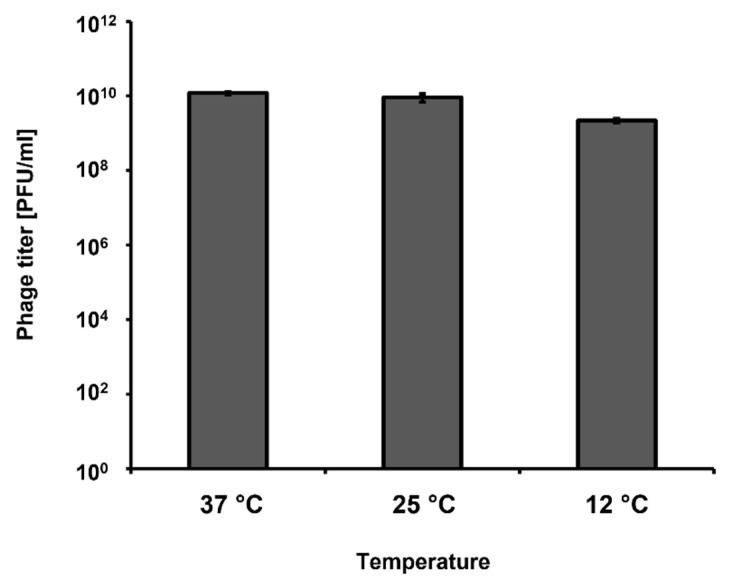
Titration of the vB_Eco4M-7 lysate using the *E. coli* O157:H7 (ST2-8624) host at different temperatures (37 °C, 25 °C and 12 °C). The titer in plaque forming units per 1 mL (PFU/mL) was determined after overnight incubation with host bacteria. Mean values from three independent experiments are shown with error bars indicating SD.

## Data Availability

Raw results are available from authors upon request.

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
