# Peer review of "A Validation System for Selection of Bacteriophages against Shiga Toxin-Producing Escherichia coli Contamination"

_toxins, 2021, doi:10.3390/toxins13090644_

Round 1
Reviewer 1 Report
This manuscript described the use of a lytic bacteriophage vB_Eco4M-7 to control the E. coli O157:H7 strain. The strain was tested on the surface of sliced cumber. The results showed that vB_Eco4M-7 can reduce the E. coli in short time but with longer treating time the phage-resistant strains were induced and eventually take over.
As the authors mentioned there is no approved antidote for STEC caused Shiga toxin intoxication. It is very important to explore the possibility of using bacteriophage as a biological control. Isolation of lytic bacteriophages specific to STEC E. coli and testing their ability of controlling the bacteria is the first step. The manuscript provided valuable information. However, in terms of the manuscript under its title I have the following major concerns:
- For controlling STEC on the vegetable surface, it is unlikely that vB_Eco4M-7 can eliminate the bacteria. The resistant strain will take over and continue release Stx so the bacteriophage vB_Eco4M-7 may not be able to control STEC O157 under field condition. More phage strains against coli O157:H7 are needed. Phage cocktail may be better option. More experiments are needed to determine if phages can be used to control STEC on vegetables.
- Because of the above reason, the title “Food protection against Shiga toxin-producing Escherichia coli contamination by using bacteriophage vB_Eco4M-7” may not be very accurate based on the presented data, a more appropriate suggested title is “A validation system for selection of bacteriophage against Shiga toxin-producing Escherichia coli contamination”.
- Shiga toxin expression or GFP expression should also be monitored during bacteriophage infection.
One additional minor comment for the manuscript:
RE: L239, it is very difficult to judge cell morphology in figure 3b. More clear pictures should be presented.
Author Response
REVIEWER’S COMMENT:
For controlling STEC on the vegetable surface, it is unlikely that vB_Eco4M-7 can eliminate the bacteria. The resistant strain will take over and continue release Stx so the bacteriophage vB_Eco4M-7 may not be able to control STEC O157 under field condition. More phage strains against coli O157:H7 are needed. Phage cocktail may be better option. More experiments are needed to determine if phages can be used to control STEC on vegetables.
RESPONSE:
We agree with this comment, and indeed it will be necessary to test more phages and perhaps different cocktails. On the other hand, this is a project for another large study. According to the next comment of the reviewer, we have modified the title of this paper, indicating that development of a validation system can be considered as the major point of this study.
REVIEWER’S COMMENT:
Because of the above reason, the title “Food protection against Shiga toxin-producing Escherichia coli contamination by using bacteriophage vB_Eco4M-7” may not be very accurate based on the presented data, a more appropriate suggested title is “A validation system for selection of bacteriophage against Shiga toxin-producing Escherichia coli contamination”.
RESPONSE:
We fully agree with this recommendation, and title of this paper has been modified, as suggested by the reviewer.
In this light, we slightly modified Abstract (lines 17-18), Introduction (lines 95-97), Discussion (lines 240-244), and Conclusions (lines :
Lines 17-18: “Here, we propose a validation system for selection of bacteriophages against STEC contamination.”
Lines 95-97: “We have developed a validation system for selection of bacteriophages against STEC contamination which was used for testing vB_Eco4M-7 and can be employed for other phages.”
Lines 240-244: “However, to properly assess usefulness of specific bacteriophages in such an application, it is necessary to develop a validation system for selection of bacteriophages against STEC contamination. In this report, we present such a new system which can be useful in choosing phages effective in preventing foodborne STEC infections.”
Lines 337-338: “A validation system for selection of bacteriophages against STEC contamination has been developed.”
REVIEWER’S COMMENT:
Shiga toxin expression or GFP expression should also be monitored during bacteriophage infection.
RESPONSE:
Again, we fully agree with the reviewer. In fact, such experiments were performed previously with phage vB_Eco4M-7, and they have already been published. Therefore, in the revised manuscript, we do mention them, and cite corresponding reference (ref. [18] – Necel et al. 2020, Sci. Rep. 10: 3743). The following text has been included (lines 279-283):
Lines 279-283: “Importantly, vB_Eco4M-7 infection of the E. coli O157:H7 host bearing a modified ST2–8624 prophage, in which the stx genes were replaced with the gfp gene, did not cause an increase in the amount of GFP, indicating that expression of Shiga toxin-encoding genes should not be activated in the presence of the tested bacteriophage [18].”
REVIEWER’S COMMENT:
One additional minor comment for the manuscript: RE: L239, it is very difficult to judge cell morphology in figure 3b. More clear pictures should be presented.
RESPONSE:
This figure has been improved, and cell morphology is well-visible now.
Reviewer 2 Report
Dear authors,
For your manuscript describing the efficiency of STEC-killing by a new bacteriophage on a sliced cucmber model (as model for microbial food protection), I like to make some suggestions to improve your work.
The following major points should be addressed:
- Explain the choice of the cucumber model and especially why it is sliced. Please refer also to the natural situation of feces-contaminated cucumbers on the peelings. Cucumbers are washed before shipping. Chlorination of washing water would be also efficient. What is the benefit of your suggested phage-killing method? The core of a cucumber has different moisture and sliced/destroyed plant cells will provide a different growth matrix for the bacteria than the peel. Please state comments on this in the discussions (maybe as advantages and limitations).
- 4 & 5 There is doubled information, delete the upper panels, and if it is needed you can use the whole width of the page for the Figures.
- Short questions on biosafety: Is the new phage usable outside a lab? Stability in the environment sufficient? Is the practicability given for the use in agriculture? How should the phages be applied in the field (application-technical)? Some of the points were shortly discussed in the manuscript. Please extend the discussion on this also for future works and applications.
Minor points:
The text has some weakness, some sentences cannot be understood as words are missing. Typos and wording errors should be corrected (yellow marks). As one example: Line 365: “positive or negative control” (without “s”). The marked words and sentences are provided in the .pdf of your manuscript. You will find there also some comments, as stated above.
Please use consistent typing of
- m.o.i: or m.o.i. or m.o.i= ? (I would prefer “MOI”)
- the degree Celsius sign sits sometimes right (superscript) and sometimes too deep
- use h instead of hrs
After correction, a proof-read by a native English speaker from the scientific field should be done to check it finally.

Author Response
REVIEWER’S COMMENT:
Explain the choice of the cucumber model and especially why it is sliced. Please refer also to the natural situation of feces-contaminated cucumbers on the peelings. Cucumbers are washed before shipping. Chlorination of washing water would be also efficient. What is the benefit of your suggested phage-killing method? The core of a cucumber has different moisture and sliced/destroyed plant cells will provide a different growth matrix for the bacteria than the peel. Please state comments on this in the discussions (maybe as advantages and limitations).
RESPONSE:
According to the reviewer’s comments, we have deeply discussed this experimental system. This discussion is included in the revised manuscript (lines 245-274), and reads as follows:
Lines 245-274: “When developing the validation system, we assessed conditions which are tighter and more endangered by bacterial infection than under field conditions of transportation and storage of vegetables. Therefore, if a tested bacteriophage can eliminate pathogenic bacteria under such experimental conditions, it should also be effective in practice. Such a system allows to avoid false positive results, where despite high efficiency of the procedure under laboratory conditions, it might be useless under conditions of real transportation and storage. To achieve this, we have used sliced cucumbers instead of whole vegetables. Colonization of whole cucumbers is considerably less likely than that of sliced ones, as the peel is a natural barrier, and it is significantly more difficult for bacteria to attach. Definitely, a sliced vegetable has different moisture and consistence which make it considerably more susceptible for colonization by bacteria than an intact one. On the other hand, we cannot exclude a partial damage of the peel during transportation and/or storage, thus, a possibility of bacterial contamination of the internal parts of cucumber is not unrealistic. Moreover, despite of procedures of washing of cucumbers before shipping, and chlorination of washing water, still, contamination with STEC cannot be excluded, especially if such procedures are not strictly followed. In fact, examples of such failures which caused STEC infections of humans have been reported [13]. It is worth reminding that for effective infection and subsequent severe symptoms in patients, a low dose of STEC (below 100 bacterial cells) is sufficient [1-6]. Considering these facts, if cucumber surface is contaminated by feces containing STEC, the peeling in a kitchen may cause a transfer of bacterial cells inside the vegetable, making their growth conditions similar to those used in our experiments. We have also used number of bacterial cells significantly higher than under natural conditions, assuming that if a tested bacteriophage is effective in eliminating its host in such an experiment, it should also be effective under field conditions. Finally, temperature of 37°C is optimal for STEC growth, thus, the use of this temperature provided fully permissive conditions for investigated bacteria. On the other hand, we have also tested ability of a bacteriophage to propagate and to lyse host cells at lower temperatures, 12 and 25°C, resembling conditions occurring during actual transportation, storage and selling. A summary for the validation system is presented in Figure 1.”
REVIEWER’S COMMENT:
4 & 5 There is doubled information, delete the upper panels, and if it is needed you can use the whole width of the page for the Figures.
RESPONSE:
Figures 4 and 5 were modified according to reviewer’s recommendations.
REVIEWER’S COMMENT:
Short questions on biosafety: Is the new phage usable outside a lab? Stability in the environment sufficient? Is the practicability given for the use in agriculture? How should the phages be applied in the field (application-technical)? Some of the points were shortly discussed in the manuscript. Please extend the discussion on this also for future works and applications.
RESPONSE:
As requested by the reviewer, we have discussed these points. Following text has been included in the revised Discussion (lines 322-335):
Lines 322-335: “Previous studies indicated that bacteriophage vB_Eco4M-7 is relatively resistant to environmental conditions [25]. It is highly stable during storage at as diverse temperatures as -20°C and 40°C, and even short time (5 min) incubation at 95°C resulted in its survival at the level of about 20% [25]. Moreover, vB_Eco4M-7 virions retained their infectivity after storage at pH 4 and 10, as well as following incubation in 0.09% SDS, 50% DMSO or chloroform [25]. This phage was also demonstrated to be resistant to various disinfectants, including 10% soap, 10% dish soap, Line Antibacterial 70, and Virusolve [18]. These properties are favorable in the light of the potential use of phage vB_Eco4M-7 in the field. However, it remains to be determined how a phage preparation should be applied. It is possible to either pour a phage lysate or spay it over the vegetables. Another option is to dip the vegetables in the solution containing bacteriophage virions. Nevertheless, the most effective way of application of bacteriophages should be identified during specific field studies, to ensure the highest efficiency of food protection against STEC contamination.”
REVIEWER’S COMMENT:
Minor points:
The text has some weakness, some sentences cannot be understood as words are missing. Typos and wording errors should be corrected (yellow marks). As one example: Line 365: “positive or negative control” (without “s”). The marked words and sentences are provided in the .pdf of your manuscript. You will find there also some comments, as stated above.
Please use consistent typing of
- m.o.i: or m.o.i. or m.o.i= ? (I would prefer “MOI”)
- the degree Celsius sign sits sometimes right (superscript) and sometimes too deep
- use h instead of hrs
After correction, a proof-read by a native English speaker from the scientific field should be done to check it finally.
RESPONSE:
We thank the reviewer for careful checking our manuscript. All the minor points were corrected in the revised version, according to reviewer’s recommendations.